# Parametrized Deep Q-Networks Learning: Playing Online Battle Arena with Discrete-Continuous Hybrid Action Space

## Abstract

Most existing deep reinforcement learning (DRL) frameworks consider action spaces that are either discrete or continuous space. Motivated by the project of design Game AI for *King of Glory* (KOG), one the world's most popular mobile game, we consider the scenario with the discrete-continuous hybrid action space. To directly apply existing DLR frameworks, existing approaches either approximate the hybrid space by a discrete set or relaxing it into a continuous set, which is usually less efficient and robust. In this paper, we propose a parametrized deep Q-network (P-DQN) farmework for the hybrid action space without approximation or relaxation. Our algorithm combines DQN and DDPG and can be viewed as an extension of the DQN to hybrid actions. The empirical study on the game KOG validates the efficiency and effectiveness of our method.

## 1 Introduction

In recent years, the exciting field of deep reinforcement learning (DRL) have witnessed striking empirical achievements in complicated sequential decision making problems that are once believed unsolvable. One active area of the application of DRL methods is to design artificial intelligence (AI) for games. The success of DRL in the game of Go (Silver et al., 2016) provides a promising methodology for game AI. In addition to the game of Go, DRL has been widely used in other games such as *Atari* (Mnih et al., 2015), *Robot Soccer* (Hausknecht & Stone, 2016; Masson et al., 2016), and *Torcs* (Lillicrap et al., 2016) to achieve super-human performances.

However, most existing DRL methods only handle the environments with actions chosen from a set which is either finite and discrete (e.g., Go and *Atari*) or continuous (e.g. *MuJoCo* and Torcs) For example, the algorithms for discrete action space include deep Q-network (DQN) (Mnih et al., 2013), Double DQN (Hasselt et al., 2016), A3C (Mnih et al., 2016); the algorithms for continuous action space include deterministic policy gradients (DPG) (Silver et al., 2014) and its deep version DDPG (Lillicrap et al., 2016).

Motivated by the applications in Real Time Strategic (RTS) games, we consider the reinforcement learning problem with a *discrete-continuous hybrid* action space. Different from completely discrete or continuous actions that are widely studied in the existing literature, in our setting, the action is defined by the following hierarchical structure. We first choose a high level action $k$ from a discrete set $\{1, 2, \cdots, K\}$; upon choosing $k$, we further choose a low level parameter $x_k \in \mathcal{X}_k$ which is associated with the $k$-th high level action. Here $\mathcal{X}_k$ is a continuous set for all $k \in \{1, \ldots, K\}$.[1] Therefore, we focus on a discrete-continuous hybrid action space

$$\mathcal{A} = \big\{(k, x_k)\big|x_k \in \mathcal{X}_k \text{ for all } 1 \leq k \leq K\big\}.$$

To apply existing DRL approaches on this hybrid action space, two straightforward ideas include:

- **Approximate $\mathcal{A}$ by an finite discrete set**. We could approximate each $\mathcal{X}_k$ by a discrete subset, which, however, might lose the natural structure of $\mathcal{X}_k$. Moreover, when $\mathcal{X}_k$ is a

---

[1]The low level continuous parameter could be deficient. It would not affect any results or derivation in this paper.

region in the Euclidean space, establishing a good approximation usually requires a huge number discrete actions.

- **Relax $\mathcal{A}$ into a continuous set**. To apply existing DRL framework with continuous action spaces, Hausknecht & Stone (2016) define the following approximate space

$$\widetilde{\mathcal{A}} = \left\{ (f_1, f_2, \ldots, f_K, x_1, x_2, \cdots, x_K) \middle| f_k \in \mathcal{F}_k, x_k \in \mathcal{X}_k \; \forall k \in [K] \right\},$$

where $\mathcal{F}_k \subseteq \mathbb{R}$. Here $f_1, f_2, \ldots, f_K$ is used to select the discrete action either deterministically (by picking $\arg\max_i f_i$) or randomly (with probability softmax$(f)$). Compared with the original action space $\mathcal{A}$, $\widetilde{\mathcal{A}}$ might significantly increases the complexity of the action space. Furthermore, continuous relaxation can also lead to unnecessary confusion by over-parametrization. For example, $(1, 0, \cdots, 0, x_1, x_2, x_3, \cdots, x_K) \in \widetilde{\mathcal{A}}$ and $(1, 0, \cdots, 0, x_1, x_2', x_3', \cdots, x_K') \in \widetilde{\mathcal{A}}$ indeed represent the same action $(1, x_1)$ in the original space $\mathcal{A}$.

In this paper, we propose a novel DRL framework, namely parametrized deep Q-network learning (P-DQN), which directly work on the discrete-continuous hybrid action space without approximation or relaxation. Our method can be viewed as an extension of the famous DQN algorithm to hybrid action spaces. Similar to deterministic policy gradient methods, to handle the continuous parameters within actions, we first define a deterministic function which maps the state and each discrete action to its corresponding continuous parameter. Then we define a action-value function which maps the state and finite hybrid actions to real values, where the continuous parameters are obtained from the deterministic function in the first step. With the merits of both DQN and DDPG, we expect our algorithm to find the optimal discrete action as well as avoid exhaustive search over continuous action parameters. To evaluate the empirical performances, we apply our algorithm to *King of Glory* (KOG), which is one of the most popular online games worldwide, with over 200 million active users per month. KOG is a multi-agent online battle arena (MOBA) game on mobile devices, which requires players to take hybrid actions to interact with other players in real-time. Empirical study indicates that P-DQN is more efficient and robust than Hausknecht & Stone (2016)'s method that relaxes $\mathcal{A}$ into a continuous set and applies DDPG.

## 2 BACKGROUND

In reinforcement learning, the environment is usually modeled by a Markov decision process (MDP) $\mathcal{M} = \{\mathcal{S}, \mathcal{A}, p, p_0, \gamma, r\}$, where $\mathcal{S}$ is the state space, $\mathcal{A}$ is the action space, $p$ is the Markov transition probability distribution, $p_0$ is the probability distribution of the initial state, $r(s, a)$ is the reward function, and $\gamma \in [0, 1]$ is the discount factor. An agent interacts with the MDP sequentially as follows. At the $t$-th step, suppose the MDP is at state $s_t \in \mathcal{S}$ and the agent selects an action $a_t \in \mathcal{A}$, then the agent observe an immediate reward $r(s_t, a_t)$ and the next state $s_{t+1} \sim p(s_{t+1}|s_t, a_t)$. A stochastic policy $\pi$ maps each state to a probability distribution over $\mathcal{A}$, that is, $\pi(a|s)$ is defined as the probability of selecting action $a$ at state $s$. Whereas a deterministic $\mu \colon \mathcal{S} \to \mathcal{A}$ maps each state to a particular action in $\mathcal{A}$. Let $R_t = \sum_{j \geq t} \gamma^{j-t} r(s_j, a_j)$ be the cumulative discounted reward starting from time-step $t$. We define the state-value function and the action-value function of policy $\pi$ as $V^\pi = \mathbb{E}(R_t | S_t = s; \pi)$ and $Q^\pi(s, a) = \mathbb{E}(R_t | S_0 = s, A_0 = a; \pi)$, respectively. Moreover, we define the optimal state- and action-value functions as $V^\pi = \sup_\pi V^\pi$ and $Q^* = \sup_\pi Q^\pi$, respectively, where the supremum is taken over all possible policies. The goal of the agent is to find a policy the maximizes the expected total discounted reward $J(\pi) = \mathbb{E}(R_0 | \pi)$, which is can be achieved by estimating $Q^*$.

### 2.1 REINFORCEMENT LEARNING METHODS FOR FINITE ACTION SPACE

Broadly speaking, reinforcement learning algorithms can be categorized into two classes: value-based methods and policy-based methods. Value-based methods first estimate $Q^*$ and then output the greedy policy with respect to that estimate. Whereas policy-based methods directly optimizes $J(\pi)$ as a functional of $\pi$.

The Q-learning algorithm (Watkins & Dayan, 1992) is based on the Bellman equation

$$Q(s, a) = \mathbb{E}_{(r_t, s_{t+1})} \big[ r_t + \gamma \max_{a' \in \mathcal{A}} Q(s_{t+1}, a') \big| s_t = s \big], \tag{2.1}$$

which has $Q^*$ as the unique solution. In the tabular setting, the algorithm updates the $Q$-function by iteratively applying the sample counterpart of the Bellman equation

$$Q(s, a) \leftarrow Q(s, a) + \alpha \big[ r_t + \gamma \max_{a' \in \mathcal{A}} Q(s', a') - Q(s, a) \big],$$

where $\alpha > 0$ is the stepsize and $s'$ is the next state observed given the current state $s$ and action $a$. However, when the state space $\mathcal{S}$ is so large that it is impossible to store all the states in memory, function approximation for $Q^*$ is applied. Deep Q-Networks (DQN) (Mnih et al., 2015) approximates $Q^*$ using a neural network $Q(s, a; w) \approx Q(s, a)$, where $w$ is the network weights. In the $t$-th iteration, the DQN updates the parameter using the gradient of the least squares loss function

$$L_t(w) = \big\{ Q(s_t, a_t; w) - \big[ r_t + \gamma \max_{a' \in \mathcal{A}} Q(s_{t+1}, a'; w_t) \big] \big\}^2. \tag{2.2}$$

In practice, DQN is trained with techniques such as experience replay (Schaul et al., 2016) and asynchronous stochastic gradient descent methods (Mnih et al., 2016) which enjoy great empirical success.

In addition to the value-based methods, the policy-based methods directly models the optimal policy. In specific, let $\pi$ be any policy. We write $p_t(\cdot|s; \pi)$ as the distribution of $S_t$ given $S_1 = s$ with actions executed according to policy $\pi$. We define the discounted probability distribution $\rho^\pi$ by

$$\rho^\pi(s') = \int_{\mathcal{S}} \sum_{t \geq 1} p_0(s) \cdot \gamma^t \cdot p_t(s'|s; \pi) \mathrm{d}s.$$

Then the objective of policy-based methods is to find a policy that maximizes the expected reward

$$J(\pi) = \int_{\mathcal{S}} \rho^\pi(s) \int_{\mathcal{A}} \pi(s, a) r(s, a) \mathrm{d}a \mathrm{d}s = \mathbb{E}_{s \sim \rho^\pi, a \sim \pi}[r(s, a)].$$

Let $\pi_\theta$ be a stochastic policy parametrized by $\theta \in \Theta$. For example, $\pi_\theta$ could be a neural network in which the last layer is a softmax layer with $|\mathcal{A}|$ neurons. The stochastic gradient methods aims at finding a parameter $\theta$ that maximizes $J(\pi_\theta)$ via gradient descent. The stochastic policy gradient theorem (Sutton et al., 2000) states that

$$\nabla_\theta J(\pi_\theta) = \mathbb{E}_{s \sim \rho^{\pi_\theta}, a \sim \pi_\theta} \big[ \nabla_\theta \log \pi_\theta(a|s) Q^{\pi_\theta}(s, a) \big]. \tag{2.3}$$

The policy gradient algorithm iteratively updates $\theta$ using estimates of (2.3). For example, the REINFORCE algorithm (Williams, 1992) updates $\theta$ using $\nabla_\theta \log \pi_\theta(a_t|s_t) \cdot r_t$. Moreover, the actor-critic methods use another neural network $Q(s, a; w)$ to estimate the value function $Q^{\pi_\theta}(s, a)$ associated to policy $\pi_\theta$. This algorithm combines the value-based and policy-based perspectives together, and is recently used to achieve superhuman performance in the game of *Go* Silver et al. (2017).

## 2.2 Reinforcement Learning Methods for Continuous Action Space

When the action space is continuous, value-based methods will no longer be computationally tractable because of taking maximum over the action space $\mathcal{A}$ in (2.2), which in general cannot be computed efficiently. The reason is that the neural network $Q(s, a; w)$ is nonconvex when viewed as a function of $a$; $\max_{a \in \mathcal{A}} Q(s, a; w)$ is the global minima of a nonconvex function, which is NP-hard to obtain in the worst case. To resolve this issue, the continuous Q-learning (Gu et al., 2016) rewrite the action value function as $Q(s, a) = V(s) + A(s, a)$, where $V(s)$ is the state value function and $A(s, a)$ is the advantage function that encodes the relative advantage of each action. These functions are approximated by neural networks $V(s; \theta^V)$ and $A(s, a; \theta^A)$, respectively, where $\theta^V$ and $\theta^A$ are network weights. The action value function is given by

$$Q(s, a; \theta^V, \theta^A) = V(s; \theta^V) + A(s, a; \theta^A).$$

Then in the $t$-th iteration, the continuous Q-learning updates $\theta^v$ and $\theta^a$ by taking a gradient step using the least squares loss function

$$L_t(\theta^V, \theta^A) = \big\{ Q(s_t, a_t; \theta^V, \theta^A) - \big[ r_t + \gamma V(s_t; \theta_t^V) \big] \big\}^2.$$

Moreover, it is also possible to adapt policy-based methods to continuous action spaces by considering deterministic policies. Let $\mu_\theta \colon \mathcal{S} \to \mathcal{A}$ be a deterministic policy. Similar to (2.3), the deterministic policy gradient (DPG) theorem (Silver et al., 2014) states that

$$\nabla_\theta J(\mu_\theta) = \mathbb{E}_{s \sim \rho^{\mu_\theta}} \left[ \nabla_\theta \mu_\theta(s) \nabla_a Q^{\mu_\theta}(s, a) \big|_{a = \mu_\theta(s)} \right]. \tag{2.4}$$

Furthermore, this deterministic version of the policy gradient theorem can be viewed as the limit of (2.3) with the variance of $\pi_\theta$ going to zero. Based on (2.4), the DPG algorithm (Silver et al., 2014) and the deep deterministic policy gradient (DDPG) algorithm (Lillicrap et al., 2016) are proposed.

## 3 RELATED WORK

**General reinforcement learning**   There is a huge body of literature in reinforcement learning, we refer readers to textbooks by Sutton & Barto (1998); Szepesvári (2010) for detailed introduction. Combined with the recent advancement of deep learning (Goodfellow et al., 2016), deep reinforcement learning becomes a blossoming field of research with a plethora of new algorithms which achieve surprising empirical success in a variety of applications that are previously considered extremely difficult and challenging.

**Finite discrete action space methods**   For reinforcement learning problems with finite action spaces, Mnih et al. (2013; 2015) propose the DQN algorithm, which first combines the deep neural networks with the classical Q-learning algorithm (Watkins & Dayan, 1992). A variety of extensions are proposed to improve DQN, including Double DQN (Hasselt et al., 2016), dueling DQN (Wang et al., 2016), bootstrap DQN (Osband et al., 2016), asynchronous DQN (Mnih et al., 2016), and averaged-DQN Anschel et al. (2017).

In terms of policy-based methods, Sutton et al. (2000) propose the REINFORCE algorithm, which is the basic form of policy gradient. An important extension is the actor-critic method (Konda & Tsitsiklis, 2000), whose asynchronous deep version A3C (Mnih et al., 2016) produces the state-of-the-art performances on the Arcade Learning Environment (ALE) benchmark (Bellemare et al., 2013).

**Continuous action space methods**   Moreover, for DRL on continuous action spaces, Silver et al. (2014) proposes the deterministic policy gradient algorithm and deterministic actor-critic algorithms. This work is further extended by Lillicrap et al. (2016), which propose the DDPG algorithm, which is an model-free actor critic algorithm using deep neural networks to parametrize the policies. A related line of work is policy optimization methods, which improve the policy gradient method using novel optimization techniques. These methods include natural gradient descent (Kakade, 2002), trust region optimization (Schulman et al., 2015), proximal gradient descent (Schulman et al., 2017), mirror descent (Montgomery & Levine, 2016), and entropy regularization (O'Donoghue et al., 2017).

**Hybrid actions**   A related body of literature is the recent work on reinforcement learning with a structured action space, which contains finite actions each parametrized by a continuous parameter. To handle such parametrized actions, Hausknecht & Stone (2016) applies the DDPG algorithm on the relaxed action space directly, and Masson et al. (2016) proposes a learning framework updating the parameters for discrete actions and continuous parameters alternately.

**Game AI**   Recently remarkable advances have been made in building AI bots for computer games using deep reinforcement learning. These games include Atari Games, a collection of video games, Texas Hold'em, a multi-player poker game, and Doom, a first-person shooter game. See Mnih et al. (2013); Heinrich & Silver (2016); Lample & Chaplot (2016); Bhatti et al. (2016) for details and see Justesen et al. (2017) for a comprehensive survey. More notably, the computer Go agent *AlphaGo* (Silver et al., 2016; 2017) achieves super-human performances by defeating the human world champion Lee Sedol. Two more complicated class of games are the real-time strategy (RTS) games and MOBA games. These are multi-agent games which involves searching within huge state and action spaces that are possibly continuous. Due to the difficulty of these problems, current research for these games are rather inadequate with most existing work consider specific scenarios instead of the full-fledged RTS or MOBA games. See, e.g., Foerster et al. (2017); Peng et al. (2017) for an recent attempt on applying DRL methods to RTS games.

## 4 PARAMETRIZED DEEP Q-NETWORKS (P-DQN)

This section introduces the proposed framework to handle the application with hybrid discrete-continuous action space. We consider a MDP with a parametrized action space $\mathcal{A}$, which consists of $K$ discrete actions each associated with a continuous parameter. In specific, we assume that any action $a \in \mathcal{A}$ can be written as $a = (k, x_k)$, where $k \in \{1, \ldots, K\}$ is the discrete action, and $x_k \in \mathcal{X}_k$ is a continuous parameter associated with the $k$-th discrete action. Thus action $a$ is a hybrid of discrete and continuous components with the value of the continuous action determined after the discrete action is chosen. Then the parametrized action space $\mathcal{A}$ can be written as

$$\mathcal{A} = \big\{ (k, x_k) \big| x_k \in \mathcal{X}_k \ \text{ for all } 1 \leq k \leq K \big\}. \tag{4.1}$$

In the sequel, we denote $\{1, \ldots, K\}$ by $[K]$ for short. For the action space $\mathcal{A}$ in (4.1), we denote the action value function by $Q(s, a) = Q(s, k, x_k)$ where $s \in \mathcal{S}$, $1 \leq k \leq K$, and $x_k \in \mathcal{X}_k$. Let $k_t$ be the discrete action selected at time $t$ and let $x_{k_t}$ be the associated continuous parameter. Then the Bellman equation becomes

$$Q(s_t, k_t, x_{k_t}) = \mathbb{E}_{(r_t, s_{t+1})} \Big[ r_t + \gamma \max_{k \in [K]} \sup_{x_k \in \mathcal{X}_k} Q(s_{t+1}, k, x_k) \Big| s_t = s \Big]. \tag{4.2}$$

Here inside the conditional expectation on the right-hand side of (4.2), we first solve $x_k^* = \operatorname{argsup}_{x_k \in \mathcal{X}_k} Q(s_{t+1}, k, x_k)$ for each $k \in [K]$, and then take the largest $Q(s_{t+1}, k, x_k^*)$. Note that taking supremum over continuous space $\mathcal{X}_k$ is computationally intractable. However, the right-hand side of (4.2) can be evaluated efficiently providing $x_k^*$ is given.

To elaborate this idea, first note that, when the function $Q$ is fixed, for any $s \in \mathcal{S}$ and $k \in [K]$, we can view

$$x_k^Q(s) = \operatorname*{argsup}_{x_k \in \mathcal{X}_k} Q(s, k, x_k) \tag{4.3}$$

as a function of state $s$. That is, we identify (4.3) as a function $x_k^Q \colon \mathcal{S} \to \mathcal{X}_k$. Then we can rewrite the Bellman equation in (4.2) as

$$Q(s_t, k_t, x_{k_t}) = \mathbb{E}_{(r_t, s_{t+1})} \Big\{ r_t + \gamma \max_{k \in [K]} Q \big[ s_{t+1}, k, x_k^Q(s_{t+1}) \big] \Big| s_t = s \Big\}.$$

Note that this new Bellman equation resembles the classical Bellman equation in (2.1) with $\mathcal{A} = [K]$. Similar to the deep Q-networks, we use a deep neural network $Q(s, k, x_k; \omega)$ to approximate $Q(s, k, x_k)$, where $\omega$ denotes the network weights. Moreover, for such a $Q(s, k, x_k; \omega)$, we approximate $x_k^Q(s)$ in (4.3) with a deterministic policy network $x_k(\cdot; \theta) \colon \mathcal{S} \to \mathcal{X}_k$, where $\theta$ denotes the network weights of the policy network. That is, when $\omega$ is fixed, we want to find $\theta$ such that

$$Q \big[ s, k, x_k(s; \theta); \omega \big] \approx \sup_{x_k \in \mathcal{X}_k} Q(s, k, x_k; \omega) \tag{4.4}$$

for each $k \in [K]$.

**Remark 4.1.** Readers who are familiar with the work by Hausknecht & Stone (2016), that also claims to handle discrete-continuous hybrid action spaces, may be curious of its difference from the proposed P-DQN. The key differences are as follows.

- In (Hausknecht & Stone, 2016), the discrete action types are parametrized as some continuous values, say $f$. And the discrete action that is actually executed is chosen via $k = \arg\max_i f(i)$. Such a trick actually turns the hybrid action space into a continuous action space, upon which the classical DDPG algorithm can be applied. However, in our framework, the discrete action type is chosen directly by maximizing the action's $Q$ value explicitly.

- The $Q$ network in (Hausknecht & Stone, 2016) uses the artificial parameters $f$ as input, which makes it an action-value function estimator of current policy ($Q^\pi$). While in our framework, the $Q$ network is actually an approximate estimator of the optimal policy's action-value function ($Q^\star$).

- We note that P-DQN is an off-policy method that can use historical data, while it is hard to use historical data in Hausknecht & Stone (2016) because there is only discrete action $k$ without parameters $f$.

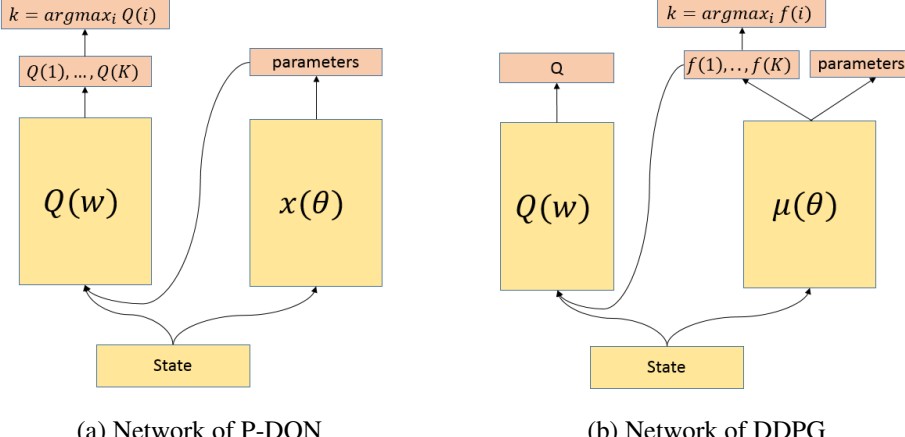

(a) Network of P-DQN          (b) Network of DDPG

Figure 1: Illustration of the networks of P-DQN and DDPG (Hausknecht & Stone, 2016). P-DQN selects the discrete action type by maximizing $Q$ values explicitly; while in DDPG, the discrete action with largest $f$, which can be seen as a continuous parameterization of $K$ discrete action types, is chosen. Also in P-DQN the state and action parameters are feed into the Q-network which outputs $K$ action values for each action type; while in DDPG, the continuous parameterization $f$, instead of the actual action $k$ taken, is feed into the Q-network.

## 5 ALGORITHM

Suppose that $\theta$ satisfies (4.4), then similar to DQN, we could estimate $\omega$ by minimizing the mean-squared Bellman error via gradient descent. In specific, in the $t$-th step, let $\omega_t$ and $\theta_t$ be the weights of the value network and the deterministic policy network, respectively. To incorporate multi-step algorithms, for a fixed $n \geq 1$, we define the $n$-step target $y_t$ by

$$y_t = \sum_{i=1}^{n-1} \gamma^i \cdot r_{t+i} + \gamma^n \cdot \max_{k \in [K]} Q\big[s_{t+n}, k, x_k(s_{t+n}, \theta_t); \omega_t\big]. \tag{5.1}$$

We define the least squares loss function for $\omega$ by

$$\ell_t^Q(\omega) = 1/2 \cdot \big\{Q\big[s_t, k_t, x_{k_t}; \omega\big] - y_t\big\}^2. \tag{5.2}$$

Moreover, since we aim to find $\theta$ that minimizes $Q[s, k, x_k(s; \theta); \omega]$ with $\omega$ fixed, we define the loss function for $\theta$ by

$$\ell_t^\Theta(\theta) = -\sum_{k=1}^{K} Q\big[s_t, k, x_k(s_t; \theta); \omega_t\big]. \tag{5.3}$$

Then we update $\omega_t$ and $\theta_t$ by gradient-based optimization methods. Moreover, the gradients are given by

$$\nabla_\omega \ell_t^Q(\omega) = \big\{Q\big[s_t, k_t, x_{k_t}; \omega\big] - y_t\big\} \cdot \nabla_\omega Q\big([s_t, k_t, x_{k_t}; \omega]\big), \tag{5.4}$$

$$\nabla_\theta \ell_t^\Theta(\theta) = -\sum_{k=1}^{K} \nabla_x Q\big[s_t, k, x_k(s_t; \theta); \omega_t\big] \cdot \nabla_\theta x_k(s_t; \theta). \tag{5.5}$$

Here $\nabla_x Q(s, k, x_k; \omega)$ and $\nabla_\omega Q(s, k, x_k; \omega)$ are the gradients of the $Q$-network with respect to its third argument and fourth argument, respectively. By (5.5) and (5.4) we update the parameters using stochastic gradient methods. In addition, note that in the ideal case, we would minimize the loss function $\ell_t^\Theta(\theta)$ in (5.3) when $\omega_t$ is fixed. From the results in stochastic approximation methods (Kushner & Yin, 2006), we could approximately achieve such a goal in an online fashion via a two-timescale update rule (Borkar, 1997). In specific, we update $\omega$ with a stepsize $\alpha_t$ that is asymptotically negligible compared with the stepsize $\beta_t$ for $\theta$. In addition, for the validity of

---

**Algorithm 1** Parametrized Deep Q-Network (P-DQN) with Experience Replay

---

**Input:** Stepsizes $\{\alpha_t, \beta_t\}_{t \geq 0}$ , exploration parameter $\epsilon$, minibatch size $B$, the replay memory $\mathcal{D}$, and a probability distribution $\mu$ over the action space $\mathcal{A}$ for exploration.

Initialize network weights $\omega_1$ and $\theta_1$.

**for** $t = 1, 2, \ldots, T$ **do**

    Compute action parameters $x_k \leftarrow x_k(s_\ell, \theta_t)$.

    Select action $a_t = (k_t, x_{k_t})$ according to the $\epsilon$-greedy policy

$$a_t = \begin{cases} \text{a sample from distribution } \mu & \text{with probability } \epsilon, \\ (k_t, x_{k_t}) \text{ such that } k_t = \arg\max_{k \in [K]} Q(s_\ell, k, x_k; \omega_t) & \text{with probability } 1 - \epsilon. \end{cases}$$

    Take action $a_t$, observe reward $r_t$ and the next state $s_{t+1}$.

    Store transition $[s_t, a_t, r_t, s_{t+1}]$ into $\mathcal{D}$.

    Sample $B$ transitions $\{s_b, a_b, r_b, s_{b+1}\}_{b \in [B]}$ randomly from $\mathcal{D}$.

    Define the target $y_b$ by

$$y_b = \begin{cases} r_b & \text{if } y_b \text{ is the terminal state} \\ r_b + \max_{k \in [K]} \gamma \cdot Q[s_{b+1}, k, x_k(s_{b+1}, \theta_t); \omega_t] & \text{otherwise.} \end{cases}$$

    Use data $\{y_b, s_b, a_b\}_{b \in [B]}$ to compute the stochastic gradient $\nabla_\omega \ell_t^Q(\omega)$ and $\nabla_\theta \ell_t^\Theta(\theta)$ defined in (5.5) and (5.4).

    Update the parameters by $\omega_{t+1} \leftarrow \omega_t - \alpha_t \cdot \nabla_\omega \ell_t^Q(\omega_t)$ and $\theta_{t+1} \leftarrow \theta_t - \beta_t \cdot \nabla_\theta \ell_t^\Theta(\theta_t)$.

**end for**

---

stochastic approximation, we require $\{\alpha_t, \beta_t\}$ to satisfy the Robbins-Moron condition (Robbins & Monro, 1951). We present the P-DQN algorithm with experienced replay in Algorithm 1.

Note that this algorithm requires a distribution $\mu$ defined on the action space $\mathcal{A}$ for exploration. In each step, with probability $\epsilon$, the agent sample an random action from $\mu$; otherwise, it takes the greedy action with respect to the current value function. In practice, if each $\mathcal{X}_k$ is a compact set in the Euclidean space (as in our case), $\mu$ could be defined as the uniform distribution over $\mathcal{A}$. In addition, as in the DDPG algorithm (Lillicrap et al., 2016), we can also add additive noise to the continuous part of the actions for exploration. Moreover, we use experience replay (Mnih et al., 2013) to reduce the dependencies among the samples, which can be replaced by more sample-efficient methods such as prioritized replay (Schaul et al., 2016).

Moreover, we note that our P-DQN algorithm can easily incorporate asynchronous gradient descent to speed up the training process. Similar to the asynchronous $n$-step DQN in Mnih et al. (2016), we consider a centralized distributed training framework where each process can compute its local gradient and synchronize with a global parameter server. In specific, each local process runs an independent game environment to generate transition trajectories and use its own transitions to compute gradients with respect to $\omega$ and $\theta$. These local gradients are then aggregated across multiple processes to update the global parameters. Note that these local stochastic gradients are independent. Thus tricks such as experience replay can be avoided in the distributed setting. Moreover, aggregating independent stochastic gradient decrease the variance of gradient estimation, which yields better algorithmic stability. We present the asynchronous P-DQN algorithm in Algorithm 2. For simplicity, here we only lay out the algorithm for each local process, which fetches $\omega$ and $\theta$ from the parameter server and computes the gradient. The parameter server stores the global parameters $\omega, \theta$ . It updates the global parameters using the gradients sent from the local processes . In addition we use the RMSProp (Hinton et al., 2012) to update the network parameters, which is shown to be more stable in practice.

## 6 GAME ENVIRONMENT: KING OF GLORY

The game *King of Glory* is a MOBA game, which is a special form of the RTS game where the players are divided into two opposing teams fighting against each other. Each team has a team *base* located in either the bottom-left or the top-right corner which are guarded by three *towers* on each of the three lanes. The *towers* can attack the enemies when they are within its attack range. Each player controls one *hero*, which is a powerful unit that is able to move, kill, perform skills, and purchase

---

**Algorithm 2** The Asynchronous P-DQN Algorithm

---

**Input:** exploration parameter $\epsilon$, a probability distribution $\mu$ over the action space $\mathcal{A}$ for exploration, the max length of multi step return $t_{\max}$, and maximum number of iterations $N_{\text{step}}$.

Initialize global shared parameter $\omega$ and $\theta$

Set global shared counter $N_{\text{step}} = 0$

Initialize local step counter $t \leftarrow 1$.

**repeat**

    Clear local gradients $d\omega \leftarrow 0$, $d\theta \leftarrow 0$.

    $t_{\text{start}} \leftarrow t$

    Synchronize local parameters $\omega' \leftarrow \omega$ and $\theta' \leftarrow \theta$ from the parameter server.

    **repeat**

        Observe state $s_t$ and let $x_k \leftarrow x_k(s_t, \theta')$

        Select action $a_t = (k_t, x_{k_t})$ according to the $\epsilon$-greedy policy

$$a_t = \begin{cases} \text{a sample from distribution } \mu & \text{with probability } \epsilon, \\ (k_t, x_{k_t}) \text{ such that } k_t = \arg\max_{k \in [K]} Q(s_t, k, x_k; \omega') & \text{with probability } 1 - \epsilon. \end{cases}$$

        Take action $a_t$, observe reward $r_t$ and the next state $s_{t+1}$.

        $t \leftarrow t + 1$

        $N_{\text{step}} \leftarrow N_{\text{step}} + 1$

    **until** $s_t$ is the terminal state **or** $t - t_{\text{start}} = t_{\max}$

    Define the target $y = \begin{cases} 0 & \text{for terminal } s_t \\ \max_{k \in [K]} Q[s_t, k, x_k(s_t, \theta'); \omega'] & \text{for non-terminal } s_t \end{cases}$

    **for** $i = t - 1, \ldots, t_{\text{start}}$ **do**

        $y \leftarrow r_i + \gamma \cdot y$

        Accumulate gradients: $d\theta \leftarrow d\theta + \nabla_\theta \ell_t^\Theta(\theta')$, $d\omega \leftarrow d\omega + \nabla_\omega \ell_t^Q(\omega')$

    **end for**

    Update global $\theta$ and $\omega$ using $d\theta$ and $d\omega$ with RMSProp (Hinton et al. (2012)).

**until** $N_{\text{step}} > N_{\max}$

---

equipments. The goal of the *heroes* is to destroy the *base* of the opposing team. In addition, for both teams, there are computer-controlled units spawned periodically that march towards the opposing *base* in all the three lanes. These units can attack the enemies but cannot perform skills or purchase equipments. An illustration of the map is in Figure 2-(a), where the blue or red circles on each lane are the *towers*. During game play, the *heroes* advance their levels and obtain gold by killing units and destroying the *towers*. With gold, the *heros* are able to purchase equipments such as weapons and armors to enhance their power. In addition, by upgrading to the new level, a hero is able to improve its unique skills. Whereas when a *hero* is killed by the enemy, it will wait for some time to reborn.

In this game, each team contains one, three, or five players. The five-versus-five model is the most complicated mode which requires strategic collaboration among the five players. In contrast, the one-versus-one mode, which is called *solo*, only depends on the player's control of a single *hero*. In a *solo* game, only the middle lane is active; both the two players move along the middle lane to fight against each other. The map and a screenshot of a *solo* game are given in Figure 2-(b) and (c), respectively. In our experiments, we play focus on the *solo* mode. We emphasize that a typical *solo* game lasts about 10 to 20 minutes where each player must make instantaneous decisions. Moreover, the players have to make different types of actions including *attack*, *move* and *purchasing*. Thus, as a reinforcement learning problem, it has four main difficulties: first, the state space has huge capacity; second, since there are various kinds of actions, the action space is complicated; third, the reward function is not well defined; and fourth, heuristic search algorithms are not feasible since the game is in real-time. Therefore, although we consider the simplest mode of *King of Glory*, it is still a challenging game for artificial intelligence.

## 7 EXPERIMENTS

In this section, we applied the P-DQN algorithm to the *solo* mode of *King of Glory*. In our experiments, we play against the default AI hero *Lu Ban* provided by the game, which is a *shooter* with long attack range. To evaluate the performances, we compared our algorithm with the DDPG algorithm (Hausknecht & Stone, 2016) under fair condition.

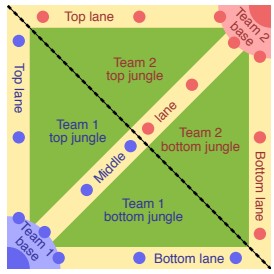 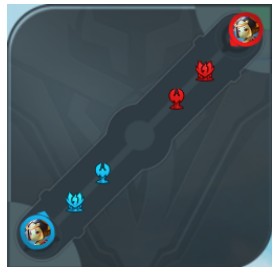 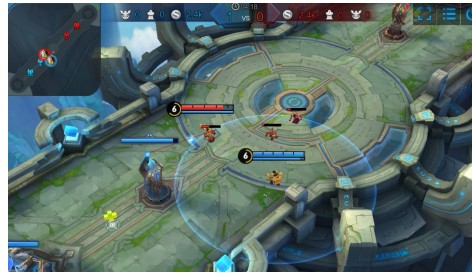

(a) The map of a MOBA game. (b) The battle field for a *solo* game. (c) A screen shot of a *solo* game.

Figure 2: (a) An illustration of the map of a MOBA game, where there are three lanes connecting two *bases*, with three *towers* on each lane for each side. (b). The map of a *solo* game of *King of Glory*, where only the middle lane is active. (c). A screenshot of a *solo* game of *King of Glory*, where the unit under a blue bar is a *hero* controlled by our algorithm and the rest of the units are the computer-controlled units.

Table 1: Action Parameters

| ActionType | Parameter | Description |
|---|---|---|
| Move | $\alpha$ | Move in the direction $\alpha$ |
| Attack | - | Attack default target |
| UseSkill1 | $(x, y)$ | Use Skill 1 at the target position $(x, y)$ |
| UseSkill2 | $\alpha$ | Use Skill 2 in the direction $\alpha$ |
| UseSkill3 | $\alpha$ | Use Skill 3 in the direction $\alpha$ |
| Retreat | - | Retreat back to our base |

## 7.1 STATE

In our experiment, the state of the game is represented by a 179-dimensional feature vector which is manually constructed using the output from the game engine. These features consist of two parts. The first part is the basic attributes of the two *heroes*, the computer-controlled units, and buildings such as the *towers* and the *bases* of the two teams. For example, the attributes of the *heroes* include *Health Point*, *Magic Point*, *Attack Damage*, *Armor*, *Magic Power*, *Physical Penetration/Resistance*, and *Magic Penetration/Resistance*, and the attributes of the *towers* include *Health Point* and *Attack Damage*. The second component of the features is the relative positions of other units and buildings with respect to the *hero* controlled by the P-DQN player as well as the attacking relations between other units. We note that these features are directly extracted from the game engine without sophisticated feature engineering. We conjecture that the overall performances could be improved with a more careful engineered set of features.

## 7.2 ACTION SPACE

We simplify the actions of a hero into $K = 6$ discrete action types: *Move, Attack, UseSkill1, UseSkill2, UseSkill3*, and *Retreat*. Some of the actions may have additional continuous parameters to specify the precise behavior. For example, when the action type is $k = Move$, the direction of movement is given by the parameter $x_k = \alpha$, where $\alpha \in [0, 2\pi]$. Recall that each *hero*'s skills are unique. For *Lu Ban*, the first skill is to throw a grenade at some specified location, the second skill is to launch a missile in a particular direction, and the last skill is to call an airship to fly in a specified direction. A complete list of actions as well as the associated parameters are given in Table 1.

## 7.3 REWARD

The ultimate goal of a *solo* game is to destroy the opponent's base. However, the final result is only available when the game terminates. Using such kind of information as the reward for training might not be very effective, as it is very sparse and delayed. In practice, we manually design the rewards using information from each frame. Specifically, we define a variety of statistics as follows. (In

the sequel, we use subscript $0$ to represent the attributes of our side and $1$ to represent those of the opponent.)

- Gold difference $GD = Gold_0 - Gold_1$. This statistic measures the difference of gold gained from killing hero, soldiers and destroying towers of the opposing team. The gold can be used to buy weapons and armors, which enhance the offending and defending attributes of the hero. Using this value as the reward encourages the hero to gain more gold.

- *Health Point* difference ($HPD = HeroRelativeHP_0 - HeroRelativeHP_1$): This statistic measures the difference of *Health Point* of the two competing heroes. A hero with higher *Health Point* can bear more severe damages while hero with lower *Health Point* is more likely to be killed. Using this value as the reward encourages the hero to avoid attacks and last longer before being killed by the enemy.

- Kill/Death $KD = Kills_0 - \text{Kills}_1$. This statistic measures the historical performance of the two heroes. If a hero is killed multiple times, it is usually considered more likely to lose the game. Using this value as the reward can encourage the hero to kill the opponent and avoid death.

- Tower/Base HP difference $THP = TowerRelativeHP_0 - TowerRelativeHP_1$, $BHP = BaseRelativeHP_0 - BaseRelativeHP_1$. These two statistics measures the health difference of the towers and bases of the two teams. Incorporating these two statistic in the reward encourages our hero to attack towers of the opposing team and defend its own towers.

- Tower Destroyed $TD = AliveTower_0 - AliveTower_1$. This counts the number of destroyed towers, which rewards the hero when it successfully destroy the opponent's towers.

- Winning Game $W = AliveBase_0 - AliveBase_1$. This value indicates the winning or losing of the game.

- Moving forward reward: $MF = x + y$, where $(x, y)$ is the coordinate of $Hero_0$: This value is used as part of the reward to guide our hero to move forward and compete actively in the battle field.

The overall reward is calculated as a weighted sum of the time differentiated statistics defined above. In specific, the exact formula is

$$
\begin{aligned}
r_t \quad = \quad & 0.5 \times 10^{-5}(MF_t - MF_{t-1}) + 0.001(GD_t - GD_{t-1}) + 0.5(HPD_t - HPD_{t-1} \\
& + KD_t - KD_{t-1} + TD_t - TD_{t-1}) + (THP_t - THP_{t-1} + BHP_t - BHP_{t-1}) + 2W.
\end{aligned}
$$

The coefficients are set roughly inversely proportional to the scale of each statistic. We note that our algorithm is not very sensitive to the change of these coefficients in a reasonable range.

## 7.4 EXPERIMENT DETAILS

In the experiments, we use the default parameters of skills provided by the game environment (usually pointing to the opponent hero's location). We found such kind of simplification does not affect to the overall performance of our agent. In addition, to deal with the periodic problem of the direction of movement, we use $(\cos(\alpha), \sin(\alpha))$ to represent the direction and learn a normalized two-dimensional vector instead of a degree (in practice, we add a normalize layer at the end to ensure this). In addition, the 6 discrete actions are not always usable, due to skills level up, lack of *Magic Point (MP)*, or skills *Cool Down(CD)*. In order to deal with this problem, we replace the $\max_{k \in [K]}$ with $\max_{k \in [K] \text{ and } k \text{ is usable}}$ when selecting the action to perform, and calculating multi-step target as in Equation 5.1.

For the network structure, recall that we use a feature vector of 179 dimensions as the state. We set both the value-network and the policy network as multi-layer fully-connected deep networks. The networks are in the same size of 256-128-64 nodes in each hidden layer, with the Relu activation function.

During the training and testing processes, we set the frame skipping parameter to 2. This means that we take actions every 3 frames or equivalently, 0.2 second, which adapts to the human reaction time, 0.1 second. We set $t_{\max} = 20$ (4 seconds) to alleviate the delayed reward. In order to encourage exploration, we use $\epsilon$-greedy sampling in training with $\epsilon = 0.255$. In specific, the first 5 type actions

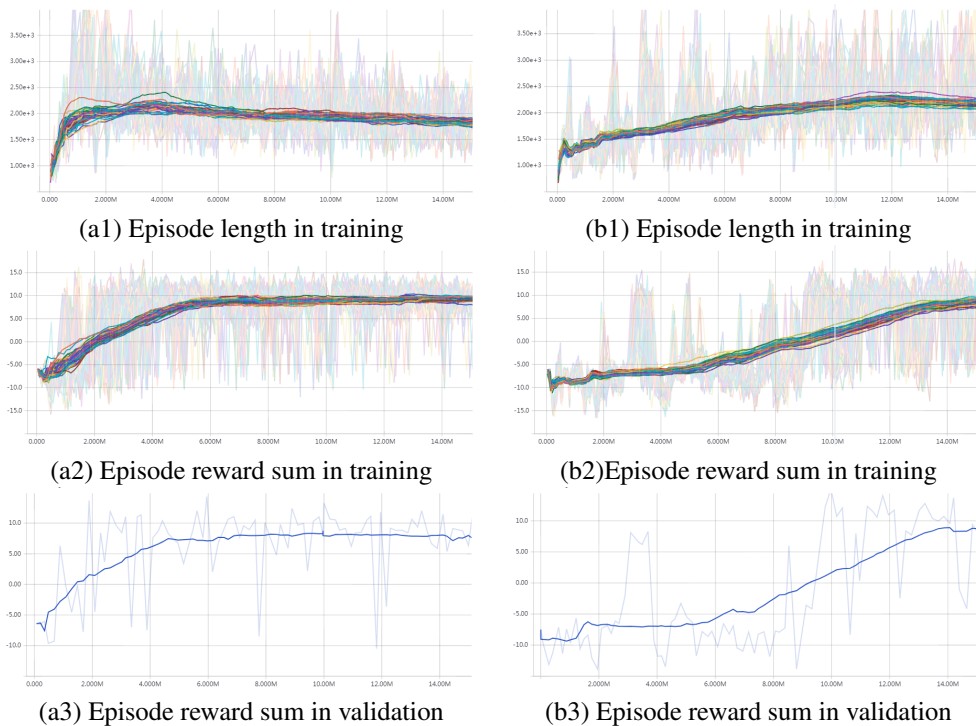

Figure 3: Comparison of P-DQN and DDPG for *solo* games with the same hero *Lu Ban*. The learning curves for different training workers are plotted in different colors. We further smooth the original noisy curves (plotted in light colors) to their running average (plotted in dark colors). In the 3 rows, we plot the average of episode lengths, reward sum averaged for each episode in training, and reward sum averaged for each episode in validation, for the two algorithms respectively. Usually a positive reward sum indicates a winning game, and vice versa. We can see that the proposed algorithm P-DQN learns much faster than its precedent work in our setting. (a) Performance of P-DQN. (b) Performance of DDPG

are sampled with probability of 0.05 each and the action "Retreat" with probability 0.005. For actions with additional parameters, since the parameters are in bounded sets, we draw these parameters from a uniform distribution. Moreover, if the sampled action is infeasible, we execute the greedy policy from the feasible ones, so the effective exploration rate is less than $\epsilon$. We uses 48 parallel workers with constant learning rate 0.001 in training and 1 worker with deterministic sampling in validation. The training and validating performances are plotted in Figure 3.

We implemented the DDPG (Hausknecht & Stone (2016)) algorithm within our learning environment to have a fair comparison. The exact network structure is plotted in Figure 1. Each algorithm is allowed to run for 15 million steps, which corresponds to roughly 140 minutes of wall clock time when paralleled with 48 workers.

From the experiments results, we can see that our algorithm P-DQN can learn the value network and the policy network much faster comparing to the other algorithm. In (a1), we see that the average length of games increases at first, reaches its peak when the two player's strength are close, and decreases when our player can easily defeat the opponent. In addition, in (a2) and (a3), we see that the total rewards in an episode increase consistently in training as well as in test settings. The DDPG algorithm may not be suitable for hybrid actions with both a discrete part and a continuous part. The major difference is that maximization over $k$ when we need to select a action is computed explicitly in P-DQN, instead of approximated implicitly with the policy network as in DDPG. Moreover, with a deterministic policy network, we extend the DQN algorithm to hybrid action spaces of discrete and continuous types, which makes the P-DQN algorithm more suitable for realistic scenarios.

## 8    CONCLUSION

Previous deep reinforcement learning algorithms mostly can work with either discrete or continuous action space. In this work, we consider the scenario with discrete-continuous hybrid action space. In contrast of existing approaches of approximating the hybrid space by a discrete set or relaxing it into a continuous set, we propose the parameterized deep Q-network (P-DQN), which extends the classical DQN with deterministic policy for the continuous part of actions. Empirical experiments of training AI for King of Glory, one of the most popular games, demonstrate the efficiency and effectiveness of P-DQN.

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
