# OpenReview forum: "PARAMETRIZED DEEP Q-NETWORKS LEARNING: PLAYING ONLINE BATTLE ARENA WITH DISCRETE-CONTINUOUS HYBRID ACTION SPACE"
_ICLR.cc/2018/Conference — Reject_

### Official Review · AnonReviewer2 · 2017-11-09
**Interesting idea but needs more analysis of results**

**Rating:** 5
**Confidence:** 4

**Review:**

This paper examines a modified NN architecture and algorithm (P-DQN) for learning in hybrid discrete/continuous action spaces. The authors come up with a clever way of modifying the architecture of parameterized-action-space DDPG (as in Hausknecht & Stone 16) in such a way that the actor only outputs values for the continuous actions and the critic outputs values for all discrete actions, parameterized by the actor’s choice of continuous actions.  Overall, I think this is an interesting and valid modification to the DDPG architecture, with results to show improved sample complexity. However, there is no quantitative analysis of why the new architecture works better, insufficient understanding of the new domain and learning task, and overall rough presentation.

Clarity: The writing clarity is rough, but understandable, with numerous minor grammar mistakes. The paper is overly long and could be improved by a more compact presentation of background, algorithms, and results.

Originality: The paper builds on DDPG and explores a novel modification to the architecture.

Significance: It’s hard to evaluate the significance of this result because of the lack of videos + information on the Moba environment. The proposed P-DQN architecture is interesting and, if the results on the Moba environment are general, could be of use in future hybrid-discrete-continuous action space domains.

Pros:
•	The modification to DDPG is genuinely interesting and does result in an algorithm that is a hybrid between DQN and DDPG.
•	The learning curves show evidence of faster learning using the P-DQN architecture.

Cons:
•	It’s difficult to confidently evaluate the merits of P-DQN vs DDPG based only on learning curves from a single, new domain. It would be nice to have explored results on additional domains such as Robot Soccer (HFO), where algorithm could have been compared directly to DDPG.
•	There is very little analysis of why P-DQN exhibits better sample complexity. The authors claim the difference stems from explicit computation over the discrete actions, but this is never analyzed.
•	Very difficult to read the axes on the plots in Fig 3.
•	Not much detail is given about the domain – who or what is the agent playing against? Is the agent playing against a bot or just learning to kill creeps? Would be great to have a video of the learned policy (or evaluation against human / scripted opponent) so that others can understand the quality of the learned policy.

---

### Official Review · AnonReviewer3 · 2017-11-27

**Rating:** 5
**Confidence:** 3

**Review:**

In this paper, the authors investigate RL agents whose action space contains discrete dimensions, and some continuous dimensions. They approach the problem by tackling the continuous dimensions with DDPG, max-marginalizing out the continuous actions, and tackling the remaining dimensions with classical Q-learning. They apply their method to a MOBA-game, King of Glory.
Methodologically, the method is a somewhat straightforward combination of DDPG and Q-learning; experimentally, they demonstrate improved performance (2-3x sample efficiency) compared to a modified DDPG algorithm from Hausknecht and Stone. Overall, methodologically, the paper is on the incremental side; experimentally, the authors attack a hard problem, and obtain moderate improvements. The most interesting part of the paper in my mind is the challenging domain of application; maybe trying their algorithm on slightly more difficult settings (different 'heroes', higher AI level) would have made the benefits of their method more evident.

Minor:
- Paper is significantly over the page limit; in many places, writing could be improved, many typos in paper
(in the first page: "project of design"-> "project of designing"; "farmework"->"framework"; "we consider the scenario" "problems that are once" are clumsy).
The used of 'parameters' for what is effectively a continuous action is a bit confusing; I realize this is borrowed from Hausknecht and Stone, but the use of the term deserves a bit more clarification (they are effectively continuous actions, but in this particular game, they parametrize a particular discrete action).
- Equation 2.2: Note that the term (r_t+ \gamma ...) is not differentiated even though it appears in the loss, various paper use different notations to denote this. As it is, the loss is slightly incorrect; same issue with the last equation on page 3.
- just after equation 2.3, the multiplier of \grad \log p_\theta for REINFORCE is not the reward r_t but the return R_t.

---

### Official Review · AnonReviewer1 · 2017-11-27
**Interesting paper but lacking in depth**

**Rating:** 4
**Confidence:** 4

**Review:**

This paper presents a new reinforcement learning approach to handle environments with a mix of discrete and
continuous action spaces. The authors propose a parameterized deep Q-network (P-DQN) and leverage learning
schemes from existing algorithms such as DQN and DDPG to train the network. The proposed loss function and
alternating optimization of the parameters are pretty intuitive and easy to follow. My main concern is
with lack of sufficient depth in empirical evaluation and analysis of the method.

Pros:
1. The setup is an interesting and practically useful one to investigate. Many real-world environments require individual actions
 that are further parameterized over a continuous space.
2. The proposed method is simple and intuitive.

Cons:
1. The evaluation is performed only on a single environment in a restricted fashion. I understand the authors are restricted in the choice of environments which require a hybrid action space. However,
 even domains like Atari could be used in a setting where the continuous parameter x_k refers to the number of
 repetitions for action k. This is similar to the work of Lakshminarayanan et al. (2017). Could you test your algorithm in such a setting?
2. Comparison of the algorithm is performed only against DDPG. Have you tried other options like PPO (Schulman et al., 2017)?
 Also, considering that the action space is simplified in the experimental setup ("we use the default parameters of skills provided by the game environment, usually pointing to
the opponent hero's location"), with only the move(\alpha) action being a hybrid, one could imagine discretizing the move
direction \alpha and training a DQN (or any other algorithms over discrete action spaces) as another baseline.
3. The reward structure seems to be highly engineered. With so many components in the reward, it is not clear
what the individual contributions are and what policies are actually learned.
4. The authors don't provide any analysis of the empirical results. Do the P-DQN and DDPG converge to the same policy?
What factor(s) contribute most to the faster learning of P-DQN? Do the values of \alpha and \beta for the two-timescale
updates affect the results considerably?
5. (minor) The writing contains a lot of grammatical errors which makes this draft below par for an ICLR paper.


Other Questions:
1. In eq. 5.3, the loss over \theta is defined as the sum of Q values over different k. Did you try other formulations of
the loss? (say, product of the Q values for instance) One potential issue with the sum could be that if some values of k dominate this sum, Q(s, k, x_k; w) might not be maximized for all k.
2. Some terms of the reward function seem to be overly dependent on historic actions (ex. difference in gold and hitpoints). This could swamp the
influence of the other terms which are more dependent on the current action a_t, which might be an issue, especially with the Markovian assumption?

References:
 Lakshminarayanan et al, 2017; Dynamic Action Repetition for Deep Reinforcement Learning; AAAI
 Schulman et al., 2017; Proximal Policy Optimization Algorithms; Arxiv

---

### Decision · Program_Chairs · 2018-01-29
**ICLR 2018 Conference Acceptance Decision**

**Decision:**

Reject

**Comment:**

The idea studied here is fairly incremental and the empirical evaluation could be improved.